# Insights, Advantages, and Barriers of Teledermatology vs. Face-to-Face Dermatology for the Diagnosis and Follow-Up of Non-Melanoma Skin Cancer: A Systematic Review

**DOI:** 10.3390/cancers16030578

**Published:** 2024-01-30

**Authors:** Georgios Nikolakis, Aristeidis G. Vaiopoulos, Ioannis Georgopoulos, Eleni Papakonstantinou, George Gaitanis, Christos C. Zouboulis

**Affiliations:** 1Departments of Dermatology, Venereology, Allergology and Immunology, Staedtisches Klinikum Dessau, Brandenburg Medical School Theodor Fontane and Faculty of Health Sciences Brandenburg, 06847 Dessau, Germany; christos.zouboulis@mhb-fontane.de; 2Docandu Ltd., London Ν8 0ES, UK; ioannis.georgopoulos@med.uoa.gr; 3Second Department of Dermatology and Venereology, “Attikon” University General Hospital, National and Kapodistrian University of Athens, 12462 Athens, Greece; avaiopoulos@gmail.com; 4Surgical Department, “Agia Sofia” Children’s Hospital, 11527 Athens, Greece; 5Private Practice of Dermatology, 19100 Attiki, Greece; elenipap2000@hotmail.com; 6Department of Skin and Venereal Diseases, Faculty of Medicine, School of Health Sciences, University of Ioannina, 45110 Ioannina, Greece; ggaitan@uoi.gr

**Keywords:** teledermatology, non-melanoma skin cancer, basal cell carcinoma, squamous cell carcinoma, diagnosis

## Abstract

**Simple Summary:**

This systematic review investigates the diagnostic concordance, advantages, and barriers of teledermatology in comparison to face-to-face dermatology for the diagnosis, management plan, and follow-up of non-melanoma skin cancer. Factors of increasing its sensitivity (teledermoscopy, quality of images, artificial intelligence, experience in generating clinical and teledermoscopy images) and its role as a tool for non-melanoma skin cancer triage for both underserved and high-risk populations are discussed.

**Abstract:**

Background: Teledermatology is employed in the diagnosis and follow-up of skin cancer and its use was intensified during and after the COVID-19 pandemic. At the same time, demographic changes result in an overall increase in non-melanoma skin cancer and skin precancerous lesions. The aim of this study was to elucidate the role of teledermatology in comparison to conventional face-to-face dermatology for such lesions and determine the advantages and limitations of this workflow for patients and physicians. Methods: Research was performed using relevant keywords in MEDLINE and CENTRAL. Relevant articles were chosen following a predetermined standardized extraction form. Results: Diagnostic accuracy and interrater/intrarater agreement can be considered comparable—although lower—than in-person consultation. Improvement of particular features such as image quality, medical history availability, and teledermoscopy can further increase accuracy. Further aspects of limitations and advantages (mean time-to-assessment, time-to-treatment, cost-effectiveness) are discussed. Conclusions: Teledermatology has comparable diagnostic accuracy with face-to-face dermatology and can be utilized both for the effective triage of non-melanocytic epithelial tumors and precancerous lesions, as well as the follow-up. Easy access to dermatologic consultation with shorter mean times to diagnostic biopsy and/or treatment coupled with cost-effectiveness could compensate for the lower sensitivity of teledermatology and offer easier access to medical care to the affected populations.

## 1. Introduction

Teledermatology (TD) utilizes telecommunication means to allow the exchange of medical information for diagnosis, consultation, treatment, and teaching [1]. The use of personal computers and smartphones allowed the unprecedented evolution of digital photography over the last decades and their implications also facilitated a revolution in telemedicine, especially during times when personal consultation was significantly restricted, such as during the COVID-19 pandemic. TD allowed the continuation of high-quality medical care while protecting more vulnerable populations and medical staff and many countries adopted a more relaxed policy concerning the exchange of medical data [1,2]. Dermatology has the advantage of being a mostly visual medical specialty and this unique characteristic makes it excellent for its implication in telemedicine [3].

Two forms of TD are currently used: the store-and-forward (SAF) technology, where clinical images are assessed asynchronously at different times and locations, and the synchronous method, where diagnosis and consultation take place simultaneously via video conferencing. The assessment can take place between physicians of different specialties (teleconsult) or directly with the patients (teleconsultation) [1].

The low density of dermatologists both in Europe and the USA in combination with their uneven distribution, the demographic alterations due to the aging populations, thus creating subpopulations with mobility restrictions but increased medical—including dermatological—needs is likely to lead to an expansion of TD use in the future in order to address relevant needs. Non-melanoma skin cancer (NMSC) is traditionally a diagnosis, which makes a high percentage of primary care providers uncomfortable with its management and might often result in direct referrals to a dermatologist [4]. Furthermore, the rates of NMSC are on a continuous rise of 2–4% [5]. Consequently, there is greater pressure on dermatologists for a timely and accurate diagnosis since early diagnosis might considerably improve patient prognosis. In this review, the barriers and advantages of TD were compared with conventional dermatologic care.

## 2. Materials and Methods

The initial search was performed on 2 October 2023 on the electronic databases CENTRAL and MEDLINE using a set of relevant search terms: “teledermatology”, “diagnosis”, “skin cancer”, “basal cell carcinoma”, “squamous cell carcinoma”, “cutaneous lymphoma”, “actinic keratosis”, “fibroxanthoma”, “dermatofibrosarcoma”, and “Merkel cell carcinoma” in order to identify all articles that included the aforementioned terms in the title or abstract of the study. The exact term combination can be found in the Appendix A. Melanoma and other non-cutaneous epithelial tumors were omitted from the retrieved results. Randomized clinical trials, case series, retrospective studies, prospective comparative studies, cross-sectional studies, cohort studies, and letters to the editor that were published after 2000 and included a comparison of TD with conventional face-to-face dermatologic consultation (FTF) for NMSC were included, as well as studies that presented certain advantages or limitations of the one method versus the other. The abstracts were scanned for eligibility by two independent reviewers (G.N. and A.V.) using a standardized and predetermined data extraction form. In cases of discrepancy, a senior author (G.G. or C.C.Z.) would decide on the eligibility of the manuscript. Only manuscripts in English and German were considered eligible. Case reports and descriptive or systematic reviews were not included in the study, although the references of the latter were screened for eligibility. A Preferred Reporting Items for Systematic Reviews and Meta-Analyses (PRISMA) chart, which summarizes the methodological approach, is demonstrated in Figure 1.

From a total of 56 articles retrieved after extraction of duplicates, 33 articles were deemed eligible and were included in the study. The results were organized into subcategories according to the barrier, limitation, or future perspective discussed and are presented below.

## 3. Results

It is crucial for the TD-FTF comparison to set the gold standard for the diagnosis of an NMSC or a non-malignant skin lesion. For the former, the reference method is histopathology, while for the latter, FTF evaluation from a dermatologist or agreed evaluation of more than one dermatologist suffices [6]. Interestingly, in the histopathological diagnosis of NMSC, a reported interrater disagreement in 2–7% of the cases is recorded, which is not taken into consideration when comparative studies between the two workflows are being assessed [7]. 

### 3.1. Teledermatology for Non-Melanoma Skin Cancer: Triage in Primary Care

One key benefit of TD is its potential as a cost-effective triage system in primary care, providing timely patient assessments (see Table 1). In a UK study, 58% of lesions were managed in primary care, saving GBP 12460 in unnecessary referrals over three years. Over 90% of patients were satisfied with the service [8]. A retrospective analysis in Sao Paulo, Brazil, with 6633 individuals over 60 years and 12,770 skin lesions showed that TD reduced the waiting time for face-to-face dermatologist assessment from 6.7 to 1.5 months, with 66% of lesions referred back to primary centers [9]. A UK comparative study revealed a significant reduction in waiting times for squamous cell carcinoma (SCC) and basal cell carcinoma (BCC) from 50 and 58 days to 28 and 35 days, respectively [10]. Similar results were found in a US retrospective chart review, where TD decreased the time to surgery from 125 to 104 days, with initial consultations completed in 4 days compared to 48 days for face-to-face referrals [11].

An Austrian observational study focused on TD’s effectiveness in diagnosing suspicious tumors preselected by general practitioners. Only 2% of lesions were non-melanocytic, and a low percentage of patients were referred for face-to-face evaluation [12].

Despite some controversial studies on efficacy, TD enhances accessibility to teledermatologists, particularly benefiting underserved populations. A retrospective cohort study showed that TD increased access to dermatologists from 11% to 44%, reducing the median waiting time to 28 days [13]. Interestingly, 11% of the cases were diagnosed as NMSC.

### 3.2. Diagnostic Concordance of TD vs. FTF: Not Such a Simple Matter

There is a considerable number of studies that address the utility of TD in the diagnosis of skin cancer, including NMSC, and evaluate the respective diagnostic accuracy (see Table 2).

A Swiss control trial randomized 30 of 78 general practitioners to an intervention, which was the TD assessment on digital images of skin lesions. Four categories were formed according to the feedback received [14], namely (1) no further investigation; (2) clinical observation; (3) biopsy; and (4) other. Diagnostic accuracy was assessed as the concordance with the 236 histological examinations, which were performed for the first three groups. One BCC was the only NMSC in the first category from TD, while no SCC, Bowen’s disease, or actinic keratosis were categorized as lesions with no further steps needed. Notably, the percentage of overall lesions in group 1 was 21.1%. The authors argue that missed malignancies in group 1, where only 10/197 histology examinations were performed, might be higher and this raises questions about the sensitivity of the method.

A prospective study from Kroemer et al. [15] found strong intrarater agreement in 104 lesions, with TD showing high specificity for 56% of NMSC cases. Another Swedish prospective multicenter study observed increased reliability with the introduction of teledermoscopy for NMSC and confirmed that a TD approach significantly reduces the time required for surgery, while 40% of the cases could have avoided a direct dermatologist appointment, thus creating slots for other patients [16].

A Hungarian review assessed TD’s diagnostic accuracy for skin cancer, achieving 85.3% concordance with FTF for primary diagnosis and 87.9% for aggregated diagnoses. Regarding the concordance of each lesion separately, the kappa coefficient was moderate for SCC (κ = 0.627), while it was higher for BCC (κ = 0.714) and actinic keratosis (κ = 0.739) [17]. A US study compared dermatologist and teledermatologist diagnostic efficacy, revealing lower agreement for non-pigmented lesions (Kappa 0.32–0.86) [18]. Lamel et al. [19] also confirmed low discordance (82% agreement) between TD and in-person dermatology, with actinic keratosis and BCC as common diagnoses. Another Brazilian study [20] focused on the diagnostic accuracy of TD in comparison to FTF dermatology and looked for the 10 most common ICD-10 diagnoses that teledermatologists have referred for a biopsy or an FTF consultation. The overall agreement with histopathology was 54%, with diagnosis of BCC showing complete agreement with histopathology, while SCC and AK showed moderate agreement. The concordance was higher if the assessment would only distinguish between malignant and benign lesions. 

### 3.3. Some Help along the Way: Introducing Teledermoscopy (TDS) to Facilitate NMSC TD Diagnosis

TDS enhances the sensitivity of TD, using digital images taken regularly either after conventional dermoscopy from dermatologists, experienced nurses, or general practitioners. In some cases, images were made by using smartphone-adapted cameras, which allow digital self-dermoscopy (See Table 3).

Zink et al. [21] compared the results of a clinical FTF examination including dermoscopy with TDS consultations including a TDS image, showing similar results for actinic keratosis and BCC, with a 92.3% agreement. The use of a derma smartphone microscope for TD diagnosis of non-melanocytic lesions had moderate interrater agreement, but weekly TD team meetings improved patient safety and reduced biopsy rates [22,23]. A retrospective study of 59,729 primary care patients revealed that a TD workflow using a dermatoscope-fitted camera, image archiving, and retrieval on a large monitor had a 9% higher probability of cancer detection compared to face-to-face referral. However, it was more time-consuming [24]. 

In a prospective observational study, TD with app-assisted teledermatologic dermoscopy expedited treatment for NMSC (BCC, SCC, or SCC in situ), reducing the median time to diagnosis and treatment to 36 days compared to 85 days in the FTF group. A direct comparison of SAF TD with and without TDS followed by FTF evaluation showed a modest (68%) diagnostic concordance for skin cancer [25].

A randomized clinical trial in Spain [26] demonstrated that adding TDS to TD improved concordance with FTF consultation from 79.20% to 94.30%, making it a cost-effective strategy for routine skin cancer screening. Incorporating digital microscopy into TD increased diagnostic accuracy by over 10%, especially for BCCs and SCCs [27] with similar sensitivity and specificity to FTF dermatologic evaluation [28,29]. However, studies suggested challenges in TDS’s diagnostic concordance, with varying agreement levels for different dermatologists and lesions. Interrater agreement was adequate for BCC (κ = 0.55–0.67) but relatively poor for invasive SCC (κ = 0.05–0.15), actinic keratosis, and SCC in situ [30]. A UK study on SAF teledermatology triage for melanoma and SCC with digital photography demonstrated a decrease in the median waiting time to clinic examination for SCC assessed with TD to 13.5 days compared to 24 days without photographs [31].

### 3.4. Teledermatology and Occupational Dermatology: Screening and Follow-Up of High-Risk Populations

TD was also applied in occupational dermatology [1]. Experience has shown that outdoor workers make scarce use of skin cancer screening programs covered by health insurance funds (see Table 3). Moreover, follow-up visits can be time-consuming for elderly patients and dermatologists. TD might improve the financial burden of such follow-ups and increase the flexibility both for the patient and the physician, thus increasing adherence.

One of the most important lesions for occupational dermatology is actinic keratosis. A large single-center comparative study was conducted in Barcelona, Spain [32] comparing TD with TD including TDS and FTF consultation for lesions such as actinic keratosis and field cancerization. A total of 1000 lesions from 636 patients were assessed by primary care physicians, followed by TD assessment with or without TDS and FTF and subsequently FTF diagnosis. It was agreed that FTF diagnosis was considered the gold standard, and if consensus was lacking or an epithelial tumor was suspected, a histopathologic evaluation was performed. TD diagnostic concordance for actinic keratosis and field cancerization were high and superior to diagnosis made by primary care physicians (92.4% vs. 62.4% and 96.7% vs. 51.8%, *p* < 0.001). TDS has significantly further increased diagnostic concordance and identification of the specific actinic keratosis subtype. The kappa coefficient for intraobserver and interobserver agreement was over 0.83, indicating the importance of such technologies for the detection of early epithelial lesions as part of primary prevention for outdoor worker screening or screening of the elderly population, which are populations expected to develop such lesions [32].

Apart from outdoor workers, a high-risk population for developing NMSC are immunosuppressed patients, such as solid organ transplant recipients or patients undergoing hemodialysis. A retrospective review of organ transplant recipients demonstrated no significant disruption in dermatologic care with the implementation of TD before and during the pandemic, showing similar rates of new NMSC diagnoses during these two periods [33].

### 3.5. Image Quality and Digital Health Innovations as a Tool for TD Improvement for Diagnosis of NMSC

Since TD is based on visual images, it is expected that the quality and standardization of the images play a major role in the results provided by TD workflows (see Table 3). An interesting prospective comparative study [34] underlined the importance of photos taken by experienced, skilled personnel, such as dermatologists or specialized nurses, for the diagnostic accuracy of TD. When clinical pictures were taken from GPs and were forwarded to teledermatologists, the diagnostic accuracy was moderate to low with a kappa coefficient of TD and histopathology diagnosis of 0.41–0.63, while FTF agreement with histopathology diagnosis was 0.55–0.73.

Another UK study independently assessed the diagnostic accuracy of 163 SAF referrals suspicious for skin cancer through an experienced dermatologist and a three-year trainee dermatologist. Interestingly, the diagnosis was identical in less than 50% of the cases and the result was independent of the experience of the doctor. The authors mention the poor quality of images or poor technique of the images taken with inadequate lighting prior to generating such images and the lack of important details of the medical history, which suggests that technical aspects and medical history details included in the referral play an equal or more important role than the experience of the clinician [35]. A retrospective comparative study from the United Kingdom [36] revealed that for 10% of the TD-assessed patients, no diagnosis was made. The authors suggested that a lack of diagnostic features, the possibility of malignancy, and technical factors might be responsible for this outcome.

In 2015, a pivotal pilot study on 20 volunteers showed that automated photography analysis based on color space transforms and morphological features achieved a modest correlation in comparison to live dermatologic consultation, with a correlation of 0.62 on the face and 0.51 on the arms [37]. The assessment of lesions suspicious for NMSC showed high interrater agreement between a mobile prevention unit physician who assessed the lesions live and two oncologists who assessed the digital images made by the former, remotely. The observers should only distinguish between malignant and benign lesions [38].

Artificial intelligence in TD was also assessed as an assisting tool in primary care for the diagnosis of skin diseases [39]. The data on skin lesions were assessed FTF in primary care and then images were taken and assessed by teledermatologists and an AI system, providing the five most probable diagnoses per image. The gold standard was the consensus of two dermatologists and if this was not the case, a third dermatologist was asked to evaluate the image separately. Mean sensitivity concerning malignant tumors was lower for the AI system both in comparison to GP and TD evaluation. Interestingly, this sensitivity was the second lowest after infectious diseases, suggesting that in the case of NMSC, AI sensitivity might not be equal to or higher than clinical evaluation, as is often reported in the literature.

## 4. Discussion

The effort of offering satisfying results on the diagnostic accuracy of TD in comparison to FTF evaluation for NMSC appears an arduous and complex procedure and the currently available data do not support solid conclusions. The available studies are particularly heterogeneous in what is established as “hit or a miss”, whether it is a conventional clinical diagnosis after FTF dermatology consultation or histology. A recent Cochrane meta-analysis [40] showed that less than 7% of malignant skin lesions were missed by TD. For lesions considered malignant by TD, usually, histology confirms the final diagnosis and is a reliable method to confirm true positive results. In contrast, negative results were usually confirmed by FTF clinical diagnosis including dermoscopy. Although biopsies for presumptive benign lesions might be considered unethical, their lack affects the reliability of excluding skin cancer. Moreover, these lesions should be followed up prospectively, to reliably exclude the development of malignancy clinically. Once again, this was not clearly stated in many studies.

From the point of view of the patient and the dermatologist, TD and TDS should ideally be characterized by high sensitivity, so that malignant lesions of patients who are not seen via a personal clinical consultation will not be missed. In contrast, regulatory authorities, which fund primary and tertiary skin cancer prevention programs, concentrate on the specificity of the method. The ability of the method to accurately assess non-malignant cases effectively reduces FTF visits, thus reducing costs for the health system and increasing slots for other patients. The meta-analysis demonstrateda sensitivity of 93.5% and specificity of 95.8% for BCC, with a surprisingly lower sensitivity despite the implementation of dermoscopic images in one study, while the FTF approach demonstrated better sensitivity and specificity [15]. For the cutaneous SCC, the sensitivity was variable, between 57–69% (decision for excision or follow-up) and 67–85% for the decision to refer a lesion. The current study confirms that newer studies and the diagnostic concordance with FTF dermatologic evaluation and histopathology remain moderate. Despite the developments and the digital innovations of recent years, which in some cases demonstrated an increase in diagnostic accuracy even in comparison to FTF, most recent studies confirmed a moderate or even low diagnostic concordance for NMSC [17,20,22,24,28] (primarily SCC, and secondarily for BCC, AK, and SCC in situ), raising concerns about the use of this workflow for this type of disease.

The visual assessment of skin lesions Is inherent to the dermatologic diagnostic process. For this reason, digital health innovations, including TD, not only facilitate the diagnostic procedure but can also be incorporated into other aspects of patient care, such as self-monitoring of skin diseases, follow-up, reminders for the application of treatment, and a more patient-oriented engagement in health choices [41]. The increasing incidence of skin cancer in our currently aging population and the presently available resource-consuming FTF screening strategies underscore the need to efficiently facilitate speedy diagnosis and/or referral procedures. TD is a rapidly growing field in this direction and its use demonstrates high patient satisfaction and confidence in the service [42], reduced waiting times, and diagnostic accuracy, which in many cases was shown to be comparable with the standard of care. The somehow moderate diagnostic concordance of TD/TDS with the reference standard might be explained by the fact that NMSC does not have the same well-established and widely accepted dermoscopic criteria for pigmented lesions. The uncertainty might lead to both misdiagnosis and easier FTF referral, in comparison to a melanocytic lesion, especially if an SCC is suspected. On the other hand, the available data show that this diagnostic inconsistency can be improved with simple measures such as the inclusion of targeted history details or the improvement of technical aspects in the acquisition of photographs. Likewise, one of the most important drawbacks in the use of TD for skin cancer screening or follow-up is the difficulty of performing a total body examination. A more lesion-based approach has similar detection rates to FTF dermatology consultation and is more time efficient.

Good clinical practice of SAF dermatology of such skin lesions includes information on the medical history of the lesion such as the site, the age of the patient, family history of skin cancer, estimated duration of the lesion, and symptoms such as bleeding, itching, and/or pain. Other risk factors such as the type of occupation, occupational or leisure time sun exposure, and immunosuppression can provide additional relevant information to support diagnostics accuracy. A concise and complete medical history form should always be an integral part of a TD/TDS consultation and to be stated in the methods section of the manuscript.

An efficient TD consultation for NMSC is shown to decrease waiting times for biopsy or diagnostic excision, offer a quick consultation for the population of rural areas, and cover the high demand for diagnosis and lesion-specific follow-up for non-melanocytic epithelial tumors and their precursors. Additionally, TD can be used to ameliorate the three most important barriers to accessing specialty dermatology care, i.e., being uninsured continuously, living in an underserved county, or being under the poverty level. For these cases, TD might be an affordable alternative for efficient NMSC screening [43].

This systematic review has limitations, primarily the difficulty of a direct head-to-head comparison of TD (with or without TDS) and FTF evaluation of NMSC because of the heterogeneity of the primary endpoints, the reference standard used, and the variability of the follow-ups, especially for cases recognized as “benign” and their documentation. Moreover, qualitative studies referring to skin cancer in general with no possibility to extract any data referring specifically to NMSC (at least as a category) might have been excluded. The databases searched for this study were MEDLINE and CENTRAL and there is a possibility that some articles in other databases (for example SCOPUS, EMBASE) could have been missed. The references of the articles retrieved were also searched, in order to expand the search and include articles not retrieved from the primary search.

## 5. Conclusions

TDS has comparable accuracy with TD for the assessment of NMSC lesions and can be a valuable tool for NMSC triage, diagnosis, and follow-up for specific lesions. Despite the controversies of a potential lower sensitivity to an FTF workflow, it is cost-effective, allows a significant rapid FTF consultation and treatment for suspicious lesions, and allows access to dermatology care for underserved and high-risk populations in a more comfortable setting. Digital innovations, tools, and technological advancements can reduce the limitations of its application and further improve the sensitivity of this workflow, thus improving diagnostic concordance with FTF consultation.

## Figures and Tables

**Figure 1 cancers-16-00578-f001:**
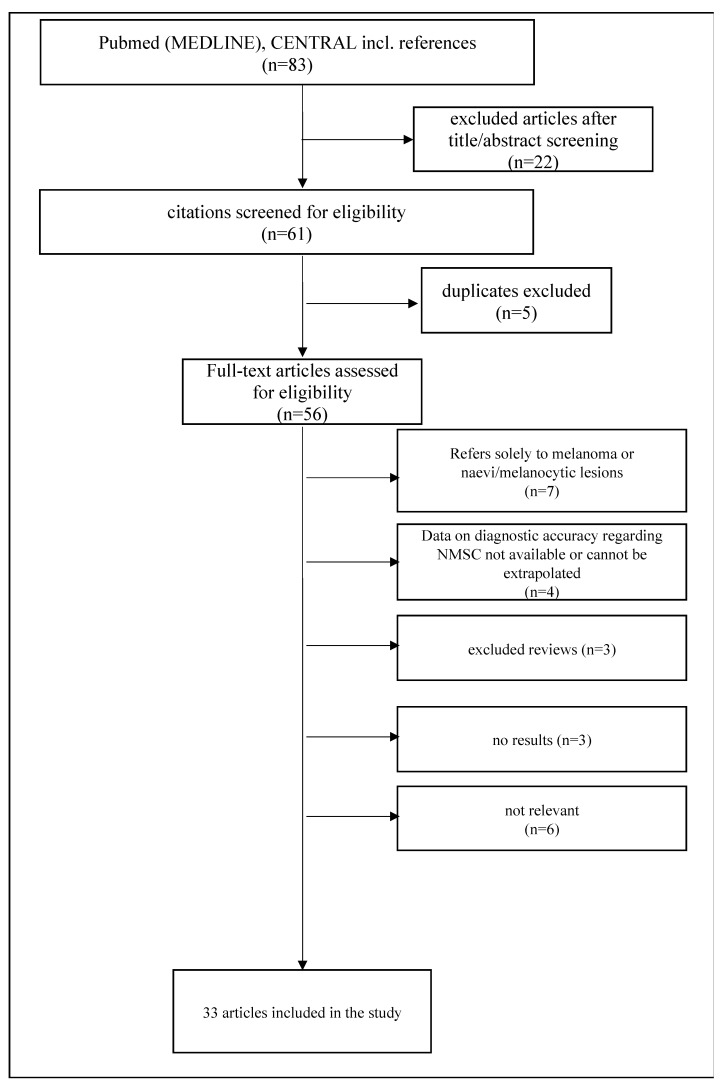
PRISMA flow chart of the study. Precancerous lesions such as Bowen’s disease and actinic keratosis were included in those articles.

**Table 1 cancers-16-00578-t001:** Studies concerning the implementation of TD as triage in primary care. AK: actinic keratosis, FTF: face-to-face, GP: general practitioner, SAF: store and forward, TD: teledermatology.

Author(s)	Country	Study Design	Study Population (n)	Intervention/Study Arms	Assessment TD Method	Reference Standard	Outcome	Potential Limitations/Bias
Livingstone et al. [8]	United Kingdom	retrospective monocentric comparative study	248 patients referred after initial GP assessment	Skin cancer diagnosis including NMSC: 102 direct referrals/FTF vs. 146 via TD	SAF: clinical photos	unknown—diagnosis from secondary care provider	TD cost-effective, timely assessment, patient satisfaction	specificity—follow-up for “benign” cases was performed exclusively by GP
Bianchi et al. [9]	Brazil	retrospective multicenter cohort study	12,770 lesions/6633 individuals	TD assessment and referral for biopsy OR FTF assessment OR return to GP for treatment	SAF: clinical photos and dermoscopy images	FTF diagnosis or histopathology if performed	2/3 of cases were returned to GP, AK between the most common diagnoses, comfortable for the elderly	specificity—follow-up for “benign” cases was performed exclusively by GP
Morton et al. [10]	United Kingdom	prospective monocentric observational study	642 lesions suspicious for skin cancer	Conventional GP referrals vs. TD consultations prior to FTF assessment/treatment	SAF: clinical photos and dermoscopy images	FTF diagnosis or histopathology if performed	TD use as triage tool, improved waiting times, reduction of the burden of FTF dermatology, photo-triage increased the sensitivity for NMSC	(-)
Hsiao et al. [11]	United States	retrospective monocentric chart review	169 patients	patients treated for skin cancer after FTF or TD assessment	SAF: not specified	histopathology	diagnostic accuracy between FTF and TD was comparable for NMSC, wait time to skin cancer surgery for TD was shorter	very specific population characteristics might not mirror the general population
Massone et al. [12]	Austria	observational multicenter study	955 lesions	TD evaluation of suspected skin cancer according to pre-trained GPs followed by referral for excision or FTF evaluation or follow-up	SAF: clinical photos and dermoscopy images	FTF diagnosis or histopathology if performed	diagnostic accuracy was 94% and the sensitivity 100%, only 1% of the TD group were referred for an FTF evaluation	(-)
Naka et al. [13]	United States	descriptive retrospective cohort study	2385 referrals	TD evaluation (44%) of suspected skin cancer from underserved US populations followed by referral for FTF evaluation or vs. direct FTF evaluation	SAF: clinical photos and dermoscopy images	FTF diagnosis or histopathology if performed	TD reduced wait times, increased primary care satisfaction, no direct head-to head comparison of diagnostic accuracy	very specific population characteristics might not mirror the general population, no data on follow-up of “benign” lesions

**Table 2 cancers-16-00578-t002:** Studies concerning TD diagnostic accuracy/concordance with the reference standard. AK: actinic keratosis, BCC: basal cell carcinoma, FTF: face-to-face, GP: general practitioner, NMSC: non-melanoma skin cancer, SAF: store and forward, SCC: squamous cell carcinoma, TD: teledermatology, TDS: teledermoscopy.

Author(s)	Country	Study Design	Study Population (n)	Intervention/Study Arms	Assessment TD Method	Reference Standard	Outcome	Potential Limitations/Bias
Tandjung et al. [14]	Switzerland	randomized control trial	979 skin lesions	TD evaluation of images made in primary care and categorization into “no further investigation”, “clinical observation”, “biopsy” and “other”.	SAF: clinical photos	FTF diagnosis or histopathology if performed	small number of avoided visits through TD, safety concerns concerning specificity of TD diagnosis (2 NMSC cases missed)	long-term data missing
Kroemer et al. [15]	Switzerland	comparative prospective study	113 skin tumors from 88 patients	clinical TD evaluation vs. dermoscopic TD evaluation of GP- and self-referrals for skin tumors and subsequent categorization to benign melanocytic, benign nonmelanocytic, malignant melanocytic and malignant nonmelanocytic lesions	SAF: clinical photos and dermoscopy images with a mobile phone camera	histopathology (malignant tumors) and FTF diagnosis (non-malignant tumors)	high concordance in differentiating benign from malignant (90%), and similar specificity of FTF in comparison to TD for NMSC, no advantages of teledermoscopy over macroscopic TD evaluation	(-)
Börve et al. [16]	Sweden	open, controlled, multicenter, prospective observational study	1562 patients	TDS evaluation via smartphone app and compatible digital microscope vs. FTF diagnosis	SAF: clinical photos and dermoscopy images	inter-rater agreement of dermatologists after FTF evaluation or histopathology	reduced waiting time from NMSC requiring surgery, increased reliability for triage through TDS, 40% of the patients could have avoided FTF	only 62% of the deemed malignant cases had a histopathologic evaluation
Jobbàgy et al. [17]	Hungary	retrospective monocentric study	749 patients with 779 lesions	TD evaluation of skin cancer lesions during the COVID-19 pandemic and categorization in 11 diagnostic groups (among them scc, BCC, AK) and triage groups followed by FTF	SAF: clinical photos	histopathology (malignant lesions) and FTF diagnosis (non-malignant lesions)	diagnostic concordance was substantial for primary (85:3%) and aggregated diagnoses (87:9%), kappa coefficient was moderate for SCC and higher for BCC	precancerous lesions (AK) were included in malignant lesions
Warshaw et al. [18]	United States	cross-sectional repeated measures equivalence study	2152 patients with 3021 lesions	TD evaluation of suspected skin cancer referrals and categorization to 1 of 17 diagnoses with up to 2 differential diagnoses, choice between 4 management plans and level of diagnostic confidence followed by FTF	SAF: clinical photos	FTF diagnosis or histopathology if performed	Diagnostic agreement had a moderate/substantial kappa of 0:32–0:86 for non-pigmented lesions incl. NMSC	male Caucasian population, teledermatologists were aware of the study, no evaluation of intra-rater reliability
Lamel et al. [19]	United States	prospective monocentric single-blind observational study	86 patients with 137 lesions	TD evaluation of lesions during a scan screening event and FTF diagnosis from another dermatologist, blinded to the TD evaluation	instant evaluation of clinical images	FTF diagnosis or histopathology if performed	substantial diagnostic agreement on primary diagnosis and management with with AK and BCC being the third and fourth most common diagnoses.	technical difficulties, no TDS
Giavina-Bianchi et al. [20]	Brazil	retrospective cohort study	30,976 patients with 55,012 lesions	TD evaluation on skin cancer (10 most frequent skin neoplasms) vs. FTF evaluation or histopathology reports with focus on diagnostic accuracy	SAF: clinical photos	FTF diagnosis or histopathology if performed	low to moderate diagnostic concordance (kappa −0:146 to 0:326) for NMSC and AK	no assessment of false negative cases

**Table 3 cancers-16-00578-t003:** Studies on teledermoscopy, cancer screening, and digital innovations. AK: actinic keratosis, BCC: basal cell carcinoma, FTF: face-to-face, GP: general practitioner, NMSC: non-melanoma skin cancer, SAF: store and forward, SCC: squamous cell carcinoma, TD: teledermatology, TDS: teledermoscopy.

Author(s)	Country	Study Design	Study Population (n)	Intervention/Study Arms	Assessment TD Method	Reference Standard	Outcome	Potential Limitations/Bias
Zink et al. [21]	Germany	prospective pilot study	26 patients	TD/TDS evaluation using a mobile phone camera and a vs. FTF diagnosis with dermoscopy	SAF: clinical photos	FTF diagnosis or histopathology if performed	diagnostic concordance in 92:3% of the cases	histology available only in 23% of the cases, FTF was performed by residents, while TD evaluation by a senior consultant
Veronese et al. [22]	Italy	retrospective observational study	144 images of suspected skin cancer lesions	FTF diagnosis using a dermatoscope vs. TDS using a dermatoscope vs. TDS using a novel smart-phone image capture device vs. TDS using the former with the interposition of a slide	SAF: clinical photos and dermoscopy images	histopathology (malignant lesions) and FTF diagnosis incl. Conventional dermoscopy with follow-up (non-malignant lesions)	TDS using conventional dermatoscopy had substantial diagnostic concordance, higher than the other two methods	retrospective character of the study
Paget et al. [23]	United Kingdom	retrospective cohort study	400 cases	TD/TDS evaluation before and aftera weekly teledermatology intradisciplinary team meeting	SAF: clinical photos and dermoscopy images	not mentioned	increase of direct discharge rate and decrease of biopsy rate after implementation of the meeting, no change in requested FTF rate	retrospective character of the study
Marwaha et al. [24]	United States	retrospective cohort study	59,729 patients	several workflows of TD/TDS evaluation of skin cancer lesions vs. direct FTF referral	SAF: clinical photos and dermoscopy images	histopathology (malignant lesions) and FTF diagnosis (non-malignant lesions)	workflow of high-resolution images with TDS had 9% higher probability of cancer detection in comparison to FTF, reduction of FTF rate by 40%, reduced wait times	potential selections bias
Bowns et al. [25]	United Kingdom	multicenter randomized control trial	208 patients	TD/TDS evaluation of GP referrals vs. FTF diagnosis	SAF: clinical photos and dermoscopy images	FTF diagnosis or histopathology if performed	modest diagnostic concordance (68%) between the two arms, sensitive but not specific	higher loss of control cases in comparison to intervention cases
Ferrándiz et al. [26]	Spain	single-center double-blind randomized control trial	454 patients	TD evaluation alone vs. TD plus TDS evaluation for suspected skin cancer, using images captured with a professional dermatoscope	SAF: clinical photos and dermoscopy images	FTF diagnosis after consulation	Diagnostic concordance with FTF increased by using TDS, increased confidence level to avoid FTF in benign lesions and was cost-effective	no data on histopathology of the lesions, regarding the reference standard
Senel et al. [27]	Turkey	retrospective monocentric cohort study	120 skin tumor lesions	TD evaluation of random benign and malignant tumors vs. TS/TDS evaluation 2 months later, conducted from the same two dermatologists vs. histopathology	SAF: clinical photos and dermoscopy images	histopathology	reliability was substantial with TD and almost perfect after TD/TDS, TDS increased diagnostic accuracy especially for SCC, BCC, actinic keratosis	TDS only with 30-fold magnification
Cheung et al. [28]	Unitred Kingdom	pilot monocentric study	76 primary care referrals of suspected skin cancer	TD evaluation vs. FTF diagnosis for single lesions suspected for skin cancer in non-hair bearing or genital sites	SAF: clinical photos	FTF diagnosis or histopathology if performed	68% of TD evaluation confident benign diagnoses were made, no FTF-assessment needed	limited size, high-rate of non-attendance for FTF diagnosis (unavailable follow-up data)
Tan et al. [29]	New Zealand	retrospective monocentric cohort study	200 patients with 491 lesions	TD/TDS evaluation of referrals in a dermatology clinic vs. FTF diagnosis	SAF: clinical photos and dermoscopy images	FTF diagnosis or histopathology if performed	TD/TDS showed approximate 100% sensitivity and 90% specificity for NMSC, 74% of TD/TDS evaluations did not need FTF evaluation	recall bias due to using the same dermatologist for TDS and FTF evaluation
Tan et al. [30]	New Zealand	retrospective study	206 patients with 979 lesions	TD/TDS evaluation for different cancerous and precancerous lesions from 5 experienced dermatologists	SAF: clinical photos and dermoscopy images	agreed TD diagnosis by all dermatologists	interrater agreement for AK/SCC in situ was moderate, moderate to very good for BCC and poor for SCC	no comparison with histopathology, selected population of elderly Fitzpatrick II patients
May et al. [31]	United Kingdom	prospective monocentric observational comparative study	43 patients with SCC out of 451 new patients	TD evaluation of melanoma and SCC vs. conventional FTF diagnosis after referral via post/fax	SAF: clinical photos	(-)	10-day decrease of waiting waiting time for SCC	sample size
Sola Ortigosa et al. [32]	Spain	prospective single-center comparative study	636 patients with 1000 keratotic skin lesions	TD±TDS evaluation vs. FTF evaluation of keratotic lesions after initial primary care assessment	SAF: clinical photos and dermoscopy images	Consensus of FTF diagnosis or histopathology (in case of disagreement)	TD: High diagnostic concordance for AK and field cancerization, further increased by TDS incl. diagnotic concordance on AK subtypes	Biopsies only for 22:5% of cases
Saranath et al. [33]	United States	retrospective medical chart review	1569 solid organ transplant recipients	TD evaluation vs. FTF evaluation of NMSC for this population during the pandemic	SAF: clinical photos	not mentioned	superior diagnostic accuracy of FTF approach than TD	gold standard not mentioned, results refer to a special population
Van der Heijden [34]	The Netherlands	prospective comparative study	76 patients	TD/TDS evaluation of lesions using images taken from GPs vs. FTF diagnosis	SAF: clinical photos and dermoscopy images	FTF diagnosis or histopathology if performed	The inter-observer reliability on diagnosis was 0:65 (substantial), the diagnostic concordance of TD/TDS with histopathology was 0:41–0:63 (moderate) and 0:90 for FTF-diagnosis	Over 1/3 of the images were reported to have bad quality
Mahendran et al. [35]	United Kingdom	prospective monocentric cohort study	163 patients	TD evaluation of suspected skin cancer GP referrals from a consultant or experienced resident vs. FTF-diagnosis from a consultant	SAF: clinical photos	FTF diagnosis or histopathology if performed	48% of the consultant’s diagnoses were identical with FTF diagnosis, less for the trainee	no statistical analysis of diagnostic agreement, recall bias possible
Mehrtens et al. [36]	United Kingdom	retrospective chart review	40,201 teleconsultations	TD evaluation of skin lesions vs. diagnosis as obtained from patient notes and histology records	SAF: clinical images and option for dermoscopic images	FTF diagnosis or histopathology if performed	10% of TD did not provide any diagnosis, diagnostic concordance with biopsied samples was 68%, BCC, AK and SCC were third to fifth most common diagnosis	retrospective character of the study
Hames et al. [37]	Australia	retrospective chart review	20 volunteers	Automatic analysis of pictures frompatients with and without actinic keratosis based on color based transforms and erythema vs. FTF approach	analysis of clinical images	TD evaluation	Correlation between of automated analysis and TD evaluation was moderate	(-)
Silveira et al. [38]	Brazil	monocentric retrospective study	416 lesions	TD evaluation of suspected skin cancer lesions by two oncologistsand classification as malignant, benign, unknown and low quality image vs. FTF approach	SAF: clinical photos	histopathology	High diagnostic accuracy (>85%) in comparison to FTF, BCC and SCC were the most common tumors	no dermnoscopic images, medical history to accompany TD missing
Escalé-Besa et al. [39]	Spain	prospective multicenter observational feasiblity study	100 patients/44 patients with a skin disease	GP evaluation of skin lesions vs. TD evaluation of GP-acquired images via smartphone camera vs. evaluation through a machine learning model	SAF: clinical photos	histopathology or concentual FTF diagnosis	diagnostic accuracy was lower for the ML model concerning the primary diagnosis and higher for the TD evl	AK was considered a benign tumor

TD: teledermatology, TDS: teledermoscopy, SCC: squamous cell carcinoma, BCC: basal cell carcinoma, AK: acitnic keratosis, FTF: face-to face, NMSC: non-melanoma skin cancer.

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
