# Peer review of "Insights, Advantages, and Barriers of Teledermatology vs. Face-to-Face Dermatology for the Diagnosis and Follow-Up of Non-Melanoma Skin Cancer: A Systematic Review"

_cancers, 2024, doi:10.3390/cancers16030578_

Round 1

Reviewer 1 Report

Comments and Suggestions for Authors

The manuscript gives an excellent overview of the published studies on teledermatology of non-melanocytic skin cancer. The results should be summarized in a Table an the text of the results should be shortened.

The authors should point out the new aspects  in comarison to  the article:

https://doi-1org-10013b5vs01df. From the cochrane library: teledermatology for diagnosing skin cancer in adults JAAD 2021

Author Response

Reviewer 1

The manuscript gives an excellent overview of the published studies on teledermatology of non-melanocytic skin cancer.

We would like to thank the reviewer for their kind remarks.

 The results should be summarized in a Table an the text of the results should be shortened.

Please also see the comments for reviewer 2. The article underwent extensive revision and the results were also added as tables according to the subcategory. This review has been conducted and presented in a way for it to be understood also by a non-expert and general reader, in order to maximize its use as a reference. There are several aspects that are discussed besides the diagnostic concordance of teledermatology with FTF dermatologic or non-dermatologic consultations, such as its use as screening in special populations, its utilization in primary care, the use of TD innovations (teledermoscopy) to improve diagnostic accuracy, which are too heterogeneous to allow an intergration in a single table and would affect the readibility of the article. For this, we decided to provide separate tables for the most important aspects  in order to satisfy the reviewers’ request. Moreover, the main text was condensed about 50%, in order to avoid an extremely lengthy manuscript.

The authors should point out the new aspects  in comarison to  the article:

https://doi-1org-10013b5vs01df [doi-1org-10013b5vs01df]. From the cochrane library: teledermatology for diagnosing skin cancer in adults JAAD 2021

The cochrane metanalysis was performed in 2018 (Chuchu et al) and assessed the diagnosis of teledermatology for skin cancer in general. This is being already mentioned in our discussion and has been updated according to the reviewer’s wish. The mentioned publication is a letter commenting on the meta-analysis.

Reviewer 2 Report

Comments and Suggestions for Authors

The authors presented a paper about "Insights, advantages and barriers of teledermatology vs. face-to-face dermatology for the diagnosis and follow-up of non-melanoma skin cancer: a systematic review".

The topic is not new but still it represents one the most frequently investigated especially during the recent COVID-19 pandemic.

I have some concerns about the reviwe as follows:

1) In the Materials and Methods sections it would be imporant to report not only the main words used to perform a systematic review but also the precise combination of terms (eg. (teledermatology) AND (diagnosis)) 

2) Since this is a systematic review the authors are supposed to have searched all of the main medical databases (such as Scopus, Cochrane library etc.) but no mention is given about that: this point should be clarified

3) What about the time period? were the articles saerched from the inception of the databases until october 2023? This point is also missing please clarify it

4) Since the number of papers included is not so elvated a table summarizing the main findings and highlighting the differences among the 30 studies included should be included

5) The discussion is poor and should be enriched with further sections at least including the limitations of the present review

Author Response

Reviewer 2

The authors presented a paper about "Insights, advantages and barriers of teledermatology vs. face-to-face dermatology for the diagnosis and follow-up of non-melanoma skin cancer: a systematic review".

We would like to thank the reviewer for the meticulous and thorough overview, which helped us improve the manuscript

The topic is not new but still it represents one the most frequently investigated especially during the recent COVID-19 pandemic.

I have some concerns about the reviwe as follows:

1) In the Materials and Methods sections it would be imporant to report not only the main words used to perform a systematic review but also the precise combination of terms (eg. (teledermatology) AND (diagnosis))

The precise combination of terms (boolean expression) has been also added as supplementary material, in order to avoid increasing the length of the manuscript. There is a reference on the supplementary material

2) Since this is a systematic review the authors are supposed to have searched all of the main medical databases (such as Scopus, Cochrane library etc.) but no mention is given about that: this point should be clarified.

The databases, which we searched for this systematic review are clearly stated in the methods  section. It is also stated, that the search was expanded using the references of the retrieved articles. This might include further articles coming from other databases. For scientific reasons, we included this aspect in the limitations of this review, summarized in the discussion part of the manuscript.

3) What about the time period? were the articles saerched from the inception of the databases until october 2023? This point is also missing please clarify it

Thank you for this very important point. The time period was included in the methods section of the manuscript: Furthermore, we also now added when did the initial search take place.

4) Since the number of papers included is not so elvated a table summarizing the main findings and highlighting the differences among the 30 studies included should be included

Please also see the reply of the comments for reviewer 1. We have performed a thorough and extended revision and created tables for each subcategry mentioned, in order to facilitate reading also for the general reader .The main drawback was the heterogeneity of the data and different outcomes which cannot be optimally depicted by one table.

5) The discussion is poor and should be enriched with further sections at least including the limitations of the present review

The heterogeneity of the studies mostly concerning what is concerned golden standard for the diagnosis of non-melanoma skin cancer was the most challenging part and has been adequately adressed in the discussion. We tried to bring further aspects, which are highlighted in the discussion, in order to improve it. Moreover, the limitations/difficulties of the study were added.

Round 2

Reviewer 2 Report

Comments and Suggestions for Authors

I have no further comments